# Can Exercise Enhance the Efficacy of Checkpoint Inhibition by Modulating Anti-Tumor Immunity?

**DOI:** 10.3390/cancers15184668

**Published:** 2023-09-21

**Authors:** Christina Brummer, Tobias Pukrop, Joachim Wiskemann, Christina Bruss, Ines Ugele, Kathrin Renner

**Affiliations:** 1Department of Internal Medicine III, Hematology and Oncology, University Hospital Regensburg, 93053 Regensburg, Germany; tobias.pukrop@ukr.de; 2Comprehensive Cancer Center Ostbayern (CCCO), 93053 Regensburg, Germany; 3Bavarian Cancer Research Center (BZKF), 93053 Regensburg, Germany; 4National Center for Tumor Diseases (NCT), Heidelberg University Hospital, 69120 Heidelberg, Germany; joachim.wiskemann@nct-heidelberg.de; 5Department of Gynecology and Obstetrics, University Medical Center Regensburg, 93053 Regensburg, Germany; christina.bruss@ukr.de; 6Department of Otorhinolaryngology, University Hospital Regensburg, 93053 Regensburg, Germany; ines.ugele@ukr.de (I.U.); kathrin.renner-sattler@ukr.de (K.R.)

**Keywords:** physical activity (PA), exercise, cancer, immune checkpoint inhibition (ICI), PD-1, PD-L1, immunotherapy, tumor microenvironment (TME), anti-tumor immunity, T cells, NK cells

## Abstract

**Simple Summary:**

Tumors escape from the host immune control by upregulation of inhibitory immune checkpoints. Immune checkpoint inhibition (ICI) has become the standard of care for many cancer entities. However, several patients do not respond to ICI because of primary or secondary resistance. Patients not benefitting from checkpoint inhibitors frequently display an immunosuppressive tumor phenotype. Combination therapy with drugs enhancing immunosurveillance improves ICI efficacy. Since physical activity can boost immune response, exercise might be a promising combinatorial therapeutic approach for ICI. Here, we review preclinical and clinical data about the impact of exercise on anti-tumor immunity and checkpoint inhibitor therapy.

**Abstract:**

Immune checkpoint inhibition (ICI) has revolutionized cancer therapy. However, response to ICI is often limited to selected subsets of patients or not durable. Tumors that are non-responsive to checkpoint inhibition are characterized by low anti-tumoral immune cell infiltration and a highly immunosuppressive tumor microenvironment. Exercise is known to promote immune cell circulation and improve immunosurveillance. Results of recent studies indicate that physical activity can induce mobilization and redistribution of immune cells towards the tumor microenvironment (TME) and therefore enhance anti-tumor immunity. This suggests a favorable impact of exercise on the efficacy of ICI. Our review delivers insight into possible molecular mechanisms of the crosstalk between muscle, tumor, and immune cells. It summarizes current data on exercise-induced effects on anti-tumor immunity and ICI in mice and men. We consider preclinical and clinical study design challenges and discuss the role of cancer type, exercise frequency, intensity, time, and type (FITT) and immune sensitivity as critical factors for exercise-induced impact on cancer immunosurveillance.

## 1. Introduction

The Food and Drug Administration’s (FDA) approval of the first checkpoint inhibitor ipilimumab, for the therapy of metastatic melanoma in 2011 has shifted the focus of anti-cancer therapy from the tumor itself towards the host’s tumor immunosurveillance [1]. The importance of the immune system for cancer control has been discussed for decades [2]. The capability of T cells to lyse tumor cells has put them into center stage in the field of immuno-oncology already years ago [3]. However, anti-tumor immune responses are complex and involve innate and adaptive immunity [4]. In general, antigen-presenting cells (APC) such as dendritic cells (DC), macrophages or B cells internalize and process tumor-associated antigens, mitigate to lymph nodes and present tumor-specific peptides by major histocompatibility complex (MHC) molecules to T cells. Co-stimulation via MHC, the T cell receptor, and other molecules, such as cluster of differentiation (CD) CD80/86 on APCs, leads to T cell activation and differentiation into T helper cells (Th) and cytotoxic T cells (CD8^+^ T). The latter can induce tumor cell apoptosis. T cell function is tightly controlled by immune checkpoints, cell surface proteins involved in immune homeostasis by activating or attenuating immune response to antigens [5]. 

Blocking of inhibitory checkpoints such as cytotoxic T-lymphocyte-associated protein 4 (CTLA-4), programmed cell death protein 1 (PD-1), programmed cell death ligand 1 (PD-L1) or lymphocyte activation gene 3 (LAG-3) can enhance the function of tumor-reactive T cells and induce anti-tumor immunity. Currently, several different immune checkpoint inhibitors targeting PD-1 (nivolumab, pembrolizumab, cemiplimab), PD-L1 (atezolizumab, avelumab, durvalumab), CTLA-4 (ipilimumab) or LAG-3 (relatlimab) are approved for cancer therapy [6]. While sometimes long-term remission under immune checkpoint inhibition (ICI) therapy can be achieved, a significant number of patients are not responding to treatment at all or relapsing upon initial response [7]. Those patients frequently display tumors with a highly immunosuppressive microenvironment [8]. 

The tumor microenvironment (TME) consists of cellular and non-cellular components, including immune cells, cancer-associated fibroblasts (CAFs), tumor endothelial cells (TEC) and the extracellular matrix (ECM). The complex crosstalk between these cell populations supports tumor growth and metastatic dissemination [9]. By expansion of immunosuppressive cells such as regulatory T cells (T_reg_), myeloid-derived suppressor cells (MDSC) or tumor-associated macrophages (TAM) [10,11,12] and concomitant contraction of immunostimulatory cells, including dendritic cells (DC), CD4 positive T helper cells (Th_1_), CD8 positive T cells (CD8^+^ T), natural killer cells (NK) and pro-inflammatory M1 like macrophages (M1), tumors suppress innate and adaptive anti-cancer mechanisms and escape from immunosurveillance [13,14,15,16,17]. Hence the cellular composition of the tumor microenvironment shapes the cytokine milieu either towards anti-tumoral, including immunostimulatory cytokines supporting antigen presentation, T cell activation and survival (IFN-γ, IL-12, IL-7, IL-15) or towards pro-tumoral fostering tumor cell proliferation, migration (IL-6, TGF-β) and suppressing immune cell activity (IL-10, IL-4) (Figure 1). 

Shifting the composition of the TME from a pro-tumoral towards an anti-tumoral landscape might improve ICI therapy [18,19]. Currently considered combinatorial approaches for ICI therapy include other immune checkpoint inhibitors, conventional chemotherapies, targeted therapies, small molecules or radiotherapy [20,21,22,23]. However, better outcomes of combination therapies are frequently associated with higher adverse effects and toxicity levels. Consequently, to enhance anti-tumor immunity, there is an ongoing effort to identify possible combination partners for ICI therapy with eligible safety profiles. 

The beneficial impact of exercise for longevity in general is well described. While exercise-induced effects on the immune system, autoimmune and other chronic diseases have already been discussed for decades, the role of exercise in cancer patients (“exercise oncology”) has only come into the spotlight over the last few years [24]. Meanwhile, physical activity is associated with a lower incidence of several types of cancer. It has been shown to improve quality of life, reduce therapy-associated side effects, and attenuate tumor-associated fatigue and cachexia [25,26,27]. Lately, there is rising evidence that exercise can also affect immune cell function in a dose-, intensity- and program-dependent matter. Since exercise is known to have immune-stimulatory effects, physical activity might boost the efficacy of immunotherapeutic approaches and represent a promising combination partner for immune checkpoint inhibitors [28,29].

## 2. Exercise-Induced Effects on the Immune System

Physical activity impacts innate and adaptive immunity and has been discussed for years [30]. However, there is an ongoing debate about underlying mechanisms [31]. Physical activity is known to exert immunomodulatory effects by exercise-induced immune cell mobilization and redistribution, exercise-mediated release of immunostimulatory myokines, exercise-induced alterations in immune cell metabolism and immunosenescence. Here, we briefly summarize exercise-induced effects on the immune system, most likely related to anti-tumor immunity. 

### 2.1. Exercise-Induced Immune Cell Mobilization and Redistribution

The transient mobilization of leukocytes into circulation during exercise is commonly reported [32,33,34,35]. This exercise-induced leukocytosis (EIL) can be observed after resistance and endurance training sessions; however, a direct comparison between different exercise programs indicates that the response tends to be more pronounced after aerobic than resistance training sessions [36,37]. EIL is driven by catecholamine-mediated downregulation of adhesion molecules, increased sheer stress and enhanced blood pressure [32,33,34,35], affecting cell populations of the innate as well as the adaptive immune system [30,38]. 

Thereby, the level of leukocyte increase seems to directly correlate with the expression of β2-adrenergic receptors on the cell surface, resulting in mobilization of mainly NK and CD8^+^ T cells [39,40,41]. For example, after high-intensity cycling (85% of Watt_max_) for 20 min, the peripheral NK cell count was five to ten-fold elevated, and CD8^+^ T cell count up to 2.5-fold [42]. Interestingly, within the NK and CD8^+^ T cell population, preferentially subsets with tissue-migrating potential and high effector functions are mobilized, resulting in enhanced elimination of neoplastic, damaged or infected cells [42,43,44,45,46,47,48]. Typically, exercise-induced leukocytosis is biphasic, peaking within 45–60 min after exercise, followed by a short-term decrease one to two hours later [49,50,51]. The observed immune cell changes are transient, and cell numbers return to pre-exercise levels usually within 24 to 48 h [52]. 

For years, this transient exercise-induced decrease in circulating immune cells has been discussed as an “immunosuppressive window” [31,53]. However, current research proposes a distinct interpretation enforcing the benefits of exercise on immune competency since leukocytes mobilized by exercise display high effector phenotypes and are redistributed towards target organs (exercise-induced redistribution, EIR) [42]. Thus, biphasic exercise-induced leukocytosis is a selective redistribution of highly effective immune cells towards peripheral tissue, e.g., mucosal surfaces, the gut, lungs and the bone marrow, resulting in enhanced immunosurveillance [31,54]. 

Alongside effects on the adaptive immune system, exercise can also result in alterations in innate immunity. Exercise furthermore increases the number of circulating monocytes, NK cells and neutrophils [38]. Moreover, exercise modulates the polarization and activation of macrophages in a tissue-specific matter [55]. Several groups have shown that exercise induces a phenotype switch from pro-inflammatory M1 like towards anti-inflammatory M2 like phenotypes, e.g., in obese or wounded mice [56,57,58]. On the contrary, exercise attenuated M2 like macrophages in other mouse experiments and promoted polarization towards the M1 like phenotype [59,60,61]. In this context, it has to be considered that recent research indicates that macrophages, including TAMs, often represent activation states not fitting into the classical M1 or M2 phenotype classification [62]. Hence, the impact of exercise on the differentiation of macrophages in different settings has to be elucidated further in future studies [55]. 

### 2.2. Exercise-Derived Immunomodulatory Myokines 

Besides exercise-induced mobilization and redistribution of immune cells as a systemic immune answer to physical activity, the skeletal muscle is regarded as a secretory and important immune regulatory organ by secretion of muscle-derived mediators, myokines [63]. In response to exercise, these myokines, such as hormones, proteins, nucleic acids and metabolites, are released and mediate the crosstalk between muscle cells and other organs [64]. 

To date, a variety of myokines has been identified, including main driver mediators of an anti-tumoral tumor microenvironment such as the interleukins (IL) IL-7, IL-15, tumor necrosis factor (TNF) and interferon-gamma (INF-γ) [24,65]. TNF and INF-γ are central in regulating T cell differentiation and function [66,67]. IL-7 and IL-15 are cytokines important for NK and T cell activation and viability [68,69,70,71]. Hence, those can improve immune responses. The first described myokine [72,73] was IL-6, exerting pleiotropic effects dependent on the target tissue [74]. In cancer patients, IL-6 is generally considered a pro-inflammatory cytokine, promoting tumor cell proliferation and metastasis [75]. Tumor-induced secretion of IL-6 fosters skeletal muscle wasting, leading to cancer cachexia [75]. Elevated IL-6 serum levels have been described as negative predictive markers for ICI therapy efficacy and patient outcome [76,77]. However, recent studies indicate that IL-6 as a myokine seems to play a more complex role in exercise and might exert contradictory and anti-tumoral effects [75]. Muscle-derived IL-6 released in response to exercise is associated with anti-inflammatory effects, including improved glucose uptake by immune cells, enhanced leukocyte mobilization and counteraction of tumor-induced muscle wasting [74,78]. Thus, the effects of IL-6 are ambivalent depending on its release as tumor-induced interleukin or exercise-derived myokine.

### 2.3. Exercise-Induced Alterations in Immune Cell Metabolism

The metabolic activity of immune cells, driven by nutrient supply and cell-intrinsic features, is closely linked to their functional profile. There is rising evidence that exercise impacts immune cell function by altering nutrient availability and directly impacting immunometabolic signaling pathways [79,80]. 

Muscles are an essential energy store, and exercise has been shown to modulate the metabolic plasma profile, including altered availability of glucose, fatty acids and glutamine [81,82,83]. Hence, physical activity impacts the nutrient supply for circulating immune cells such as lymphocytes [83]. Exercise triggers glucose and glutamine consumption in lymphocytes [84]. This metabolic reprogramming affects their functional status, resulting in enhanced expression of IL-2 and decreased expression of IL-4. Since glutamine is an indispensable metabolite for T lymphocytes and muscle cells are the main source, glutamine might become limiting in cancer patients suffering from cachexia [81,85,86]. Exercise might also improve glutamine supply to immune cells [87].

Furthermore, training increases mitochondrial biogenesis in muscles and has also been shown to affect mitochondrial mass in lymphocytes, rendering them more resistant to TME [88,89,90]. Moreover, exercise has a beneficial effect on glucose homeostasis and insulin levels, ameliorating chronic inflammatory diseases such as diabetes, related to a higher risk for cancer development [81,91,92]. Exercise reduces the percentage of body fat and, thereby, the level of circulating free fatty acids and secreted adipokines, supporting rather pro-inflammatory immune cell populations [93]. On the other hand, physical activity can also result in increased muscle-derived lactic acid release, known to blunt immunosurveillance [94]. However, as exercise-induced increase in plasma lactic acid is rapidly buffered in contrast to the TME [94,95], muscle-derived lactate does not lead to significant plasma acidification. It has no hostile impact on circulating immune cells [96]. 

Besides effects on metabolites and nutrient supply, exercise directly impacts intracellular signaling cascades and transcription factors playing an important role in metabolic regulation, such as adenosine monophosphate-activated protein kinase (AMPK), mammalian target of rapamycin (mTOR) and hypoxia-inducible factor 1-alpha (HIF1α) [97,98,99,100,101,102,103,104]. Furthermore, contraction-associated secretion of myokines is known to reprogram immune cell metabolism, e.g., of macrophages: Exercise-derived IL-6 and IL-10 have been shown to increase oxidative metabolism in macrophages [105,106,107,108]. In macrophages, metabolic reprogramming is directly linked to activation and polarization; a shift towards increased oxidative metabolism is generally known to promote the anti-inflammatory macrophage phenotype [79]. However, whether an exercise-induced switch in macrophage polarization can also influence tumor-associated macrophages in the TME and eventually result in anti- or pro-tumoral effects is still under research, as discussed in Section 2.1 [55,109]. 

### 2.4. Exercise-Mediated Effects on Immunosenescence 

Moreover, favorable effects of exercise on the age-related deterioration of the immune system, described by the term immunosenescence, have been reported lately [110]. Aging profoundly affects immune cell populations and lymphoid organs, including thymus degeneration, reduced T cell output and remodeling of T cell immunity. This results in the accumulation of senescent and exhausted immune cell phenotypes characterized by altered mitochondrial function, the loss of co-stimulatory molecules and lowered cytokine production [111,112,113]. Immunosenescence leads to an increase in autoimmune diseases, infections, reduced vaccination response and higher tumor incidence in the elderly [114,115]. 

Regular physical activity can reverse and prevent this age-associated decline in immune competence [110]. First evidence was delineated from vaccination studies showing that regular physical activity leads to higher levels of vaccination response compared to sedentary adults [116,117]. Later, it was shown that exercise supports thymopoetic output probably by muscle-secreted IL-7 [118,119]. Additionally, physical activity is known to reduce senescent and exhausted CD8^+^ T cells, fostering the production of more immunologically responsive T cells [120]. 

Altogether, exercise-induced leukocyte trafficking, secretion of myokines and reversion of immunosenescence support the role of physical activity as a potential candidate for immunotherapeutic approaches in anti-cancer therapy [24,31,110].

## 3. Exercise-Induced Impact on the TME and Tumor Immunity

There is broad data on the growth-inhibitory effect of exercise on different tumor entities [121,122]. However, underlying mechanisms are not fully understood and are still under investigation. The crosstalk between muscle, immune and cancer cells is highly complex and exercise-induced tumor growth retardation is probably a result of an interplay between multiple factors influencing the TME [24]. Thereby, activity-induced myokine production, tumor vascularization and blood flow control alterations, and changes in cancer cell metabolism and immune regulation are discussed to contribute to beneficial exercise-induced effects on cancer cell growth [25,27,123] (Figure 2). 

A detailed overview of exercise-induced effects on the TME was provided by Koelwyn and colleagues recently [109]. In this review, we focus on exercise-induced effects on anti-tumor immunity. However, since intra-tumoral immune cell infiltration and function are known to closely interplay with other components of the TME, such as tumor metabolism and vascularization and vice versa, we will first give a short overview of exercise-induced effects on the TME in general. 

Briefly, solid tumors display abnormal tumor vascularization, leading to impaired immune infiltration, restricted oxygen supply and, consequently, intra-tumoral hypoxia. Hypoxia induces HIF-1α, promoting cancer invasion, metastasis and altered intra-tumoral glucose metabolism, resulting in lactate accumulation and concomitant acidification [124]. Highly glycolytic tumor areas are known to be hostile to immune effector functions. Reducing tumor glycolysis by suppressing lactate dehydrogenase A (LDH-A) and the concomitant acidification of the TME have been shown to increase T and NK cell abundance and activity while reducing the number of MDSC. This leads to improved tumor growth control [94]. Thus, normalizing intra-tumoral blood flow and reducing hypoxic areas could alter the metabolic tumor profile and enhance anti-tumor immunity. In several tumor mouse models exercise has been shown to normalize tumor vascularization and promote anti-tumoral immune cell infiltration and drug delivery [125,126,127]. Different underlying mechanisms are discussed, including redistribution of cardiac output and exercise-induced vascular sheer stress leading to secretion of angiogenesis mediators such as vascular endothelial growth factor A (VEGF-A), macrophage inflammatory protein 1α (MIP1α) and nitric oxide (NO) [126,128,129]. 

Besides exercise-induced effects on angiogenesis, physical activity can also directly lower tumor lactate metabolism by fostering oxygen supply and reducing the expression of LDH-A and basigin, the chaperon for the lactate-transporting monocarboxylate transporter (MCT) [130]. In line, a study in Wistar tumor-bearing rats showed an exercise-triggered shift towards glucose oxidation in macrophages and lymphocytes related to better survival in tumor bearing trained rats [131]. Those data are in accordance with studies showing better anti-tumor efficacy of immune cells displaying increased oxidative phosphorylation capacity [85,132,133,134]. 

Exercise is a promising therapeutic approach to remodel the tumor microenvironment by targeting tumor vascularization, metabolism, and immunity. In the following section, we will discuss pre-clinical and clinical studies about exercise-induced effects on anti-tumor immunity in terms of training modalities in detail.

### 3.1. Preclinical Data 

Most exercise experiments in rodents are conducted in syngeneic transplanted mouse models, and few in drug-induced or transgenic tumor models. Due to the complex biology of carcinogenesis, it is important to discuss exercise-induced effects in the context of tumor inoculation. In transplanted tumor models (=subcutaneous or orthotopic implantation of tumor cells), exercise interventions can be grouped into preventive versus therapeutic settings. In a preventive setting, exercise starts before tumor inoculation, and in a therapeutic setting, after tumor inoculation. In contrast to transplanted tumor mouse models, for exercising mice with drug-induced or transgenic tumors, the determination of exercise timing in relation to tumor initiation is more difficult. It is most likely up to a mixture of preventive and therapeutic settings that can hardly be transferred to humans (Figure 3). Therefore, this review does not include a few studies of exercise-induced effects on anti-tumor immunity in transgenic or drug-induced tumor models. 

The broadest evidence for exercise-induced effects on anti-tumor immunity is reported for breast cancer mouse models such as 4T1 or E0771. 4T1 resembles a spontaneously metastatic poorly immunogenic triple-negative breast cancer (TNBC) lacking estrogen receptor (ER), progesterone receptor (PR) and epidermal growth factor receptor 2 (HER2) [135,136]. E0771 is a breast cancer model derived from a C57BL/6 mouse that some authors classify as ER-positive, some as TNBC [137]. Further studies about exercise-induced impact on anti-tumor immunity are reported for melanoma (B16-F10), pancreatic ductal adenocarcinoma (KPC4662), undifferentiated carcinoma (Ehrlich cells), lung (Lewis lung cancer) or liver cancer (Hepa1-6). Studies are reviewed in detail in the following section. Nearly all exercise interventions in rodents represent aerobic activities such as swimming or running (wheel, treadmill). Aerobic exercise programs are performed on a programmed (=controlled access to sports equipment, e.g., running wheel) or voluntary basis (=unlimited access). Data about the effect of resistance training on anti-cancer immunity and ICI in mice is rare. To our knowledge, only one study applied a combined endurance and strength workout (5 x/week, each 40–60 min) consisting of 30–40 min treadmill running followed by dynamic (repetitive climbing on top of an inverted screen) and static (hanging on a metal cloth hanger with forepaws) strength training [138]. However, since immunodeficient NOD-SCID mice were used, results hardly represent a transferable model for studying anti-tumor immunity and, therefore, are not included in this review. Since cancer type and exercise program significantly impact exercise-induced effects, we review data according to cancer types and exercise programs rather than from an immunological point of view in contrast to other groups [29,139,140,141,142,143]. Thereby it is important to distinguish whether exercise interventions are conducted in a preventive setting (=starting exercise before tumor inoculation, Table 1) or in a therapeutic setting (=starting exercise after tumor inoculation, Table 2).

#### 3.1.1. Preventive Setting

Programmed exercise: In a preventive setting (=starting exercise before tumor inoculation), swimming (1 h/d, 50% of max. capacity, five d/week) four weeks before inoculation of undifferentiated carcinoma cells (Ehrlich tumor) could lower levels of intra-tumoral macrophages and neutrophils and reduce tumor growth [144]. In another study on hepatocellular carcinoma, short sessions of swimming (5–8 min/d, five x/week) before tumor inoculation significantly delayed tumor growth, reduced lung metastasis and prolonged median overall survival in comparison to sedentary control groups (control 59 d vs. swimming 68 d). The observed effects were associated with reduced levels of intra-tumoral T_reg_ infiltration [145]. For breast cancer, two studies (Hagar et al. and Wang et al.) examined the impact of programmed running in the 4T1 mouse model [146,147]. Both groups showed that pretraining in running wheels for several weeks (5 d/week, programmed) before tumor inoculation significantly slowed down tumor progression compared to sedentary mice. Effects were mediated by improved anti-tumor immune response, including enhanced levels of intra-tumoral NK cells and a significantly higher CD8^+^ T/T_reg_ ratio. Interestingly, exercise-induced delay in tumor growth could not be reproduced in athymic BALB/c mice lacking mature T cells [146]. Wang et al. (three weeks of pretraining, five d/week) reported significant results only at higher volume and intensity (10–15 m/min, distance 600–900 m/d). At distances of 360 m/d at a velocity of 6 m/min, no effects were observed [147]. In contrast, Hagar et al. (eight weeks of pretraining, five d/week) showed significant exercise-induced effects already at short overall daily running distances (250 m/d); however, they conducted exercise experiments at a higher intensity per session (12 m/min) [146]. These results indicate a dose-dependency of exercise-induced effects on anti-tumor immunity. 

Voluntary exercise: Data on tumor growth retardation by voluntary exercises is more contradictory. Turbitt et al. and Garritson et al. investigated the impact of wheel running starting before breast cancer inoculation (six to eight weeks) in the same tumor model as Wang et al. (4T1), but voluntarily. In contrast to programmed exercise, voluntary running could not retard tumor growth (Turbitt and Garritson) and led to only slight changes in anti-tumoral immunity (CD8^+^ T ↑, MDSC ↓) [148,149]. For another breast cancer mouse model (E0771), neither growth inhibition nor changes in intra-tumoral immune cell composition of voluntarily running before tumor inoculation have been reported [150]. In contrast, Rundqvist et al. showed significantly enhanced intra-tumoral CD8^+^ T cell levels and growth retardation for mice that started running two weeks before inoculation, also in a voluntary setting (I3TC). 

These results indicate that besides intensity and timing, the exercise modality (programmed vs. voluntary) plays a significant role in modulatory effects on anti-tumor immunity, which we discuss in detail in the following section. Similar results were reported by Pedersen et al., who examined the impact of voluntary wheel running in B16-F10 melanoma and Lewis lung cancer [151]. In mice trained before inoculation of B16 melanoma cells, tumor growth was reduced by up to 60% compared to the sedentary control group. Exercise-induced growth inhibition was associated with significantly higher levels of intra-tumoral CD4^+^ T cells, CD8^+^ T cells, NK cells and DCs, indicating a shift towards an anti-tumor microenvironment. Furthermore, the expression of key immune regulatory molecules (e.g., PD-L1, PD-L2, PD-1) was upregulated [150].

Interestingly, in this tumor model, the growth-inhibitory effect of exercise seemed to be NK cell but not T cell-mediated since results were similar in athymic mice lacking functional T cells, whereas antibody-mediated depletion of NK cells abolished the growth inhibition. Mechanistically, reprogramming of the TME was mediated by exercise-derived epinephrin and IL-6 since propranolol and anti-IL6 treatment blunted exercise-induced intra-tumoral NK and T cell infiltration and growth inhibition. While epinephrin alone could mimic the exercise-induced alteration of intra-tumoral immune cells and induce growth inhibition, monotherapy with IL-6 could not. This indicates that epinephrin-mediated cell mobilization to the blood is required before IL-6-mediated redistribution of immune cells into the TME. This aligns with previously discussed exercise-induced effects on the immune system (Section 2.1). 

#### 3.1.2. Therapeutic Setting

Programmed exercise: In a pancreatic ductal adenocarcinoma mouse model, a post-implant exercise program restricted pancreatic tumor growth by remodeling the anti-tumor immunity [152]. Programmed treadmill running sessions (5 d/week, 9 m/min, 270 m/d) led to significantly reduced levels of intra-tumoral MDSCs and accumulation of CD8^+^ T cells in mice transplanted with KPC4662 cells. Subset analysis showed an exercise-induced shift of T lymphocyte composition from exhaustive towards effector phenotypes. Interestingly, no increase in CD8^+^ T cell function in the tumor-draining lymph node or spleen has been observed, indicating a tumor-site-specific effect. Since exercise-induced effects were lost in athymic mice, the impact on TME composition seems mainly mediated by CD8^+^ T cells. In line with the results of Pedersen et al., exercise-induced growth retardation was mediated by epinephrin-induced mobilization of T cells to the blood pool (exercise-induced leukocytosis) followed by myokine-mediated redistribution towards the TME (here: IL-15). Vice versa, propranolol treatment abrogated the exercise-induced mobilization of CD8^+^ T cells and growth inhibition. 

Similar results are reported for other cancer entities. Gomes-Santos et al. showed that seven days of exercise post-tumor inoculation at moderate intensity (60% of Vmax for 30–45 min per day, daily) could significantly reduce tumor growth in three different syngeneic transplanted breast cancer mouse models (E0771, MCa-M3C, EMT6). Exercise-induced growth retardation was mediated by increased intra-tumoral CD8^+^ T cell infiltration, while no significant changes in T_reg_ cells, tumor-associated macrophages, DCs or NK cells were found [153]. Significant growth-inhibitory impact of moderate training (5 d/week, 18 m/min, 540 m/d) associated with increased intra-tumoral NK and CD8^+^ T cells and decreased MDSCs has also been reported by Wennerberg et al. for another breast cancer mouse model (4T1) [154].

Voluntary exercise: In contrast to the detected effects of programmed running, post-transplantation exercise on a voluntary basis did not result in reprogramming of TME composition or tumor growth retardation neither in breast cancer (E0771, [155]) nor melanoma (B16-F10, [151,155]) mouse model. In contrast to mice pretrained before tumor inoculation, Pedersen et al. showed that mice starting exercise after tumor injection (d0 to d14) did not reveal any significant reduction in tumor size compared to the sedentary group. These results agree with the observations of other groups. Buss et al. reported that voluntary wheel running post-tumor implantation (d0–17) neither slowed tumor growth rate nor altered numbers of intra-tumoral NK cells, CD8^+^ T or T_reg_ cells in B16-F10 melanoma and E0771 breast cancer [155,156]. 

**Table 1 cancers-15-04668-t001:** Preclinical data on the exercise-induced impact on anti-tumor immunity in a preventive setting (=starting exercise before tumor cell inoculation). Day 0 (d0) is the time point tumor cells were transplanted. Arrows upwards (↑) indicate an increase, arrows downwards (↓) indicate a decline/reduction in tumor growth or intra-tumoral immune cell numbers.

Entity	Study	Cell line	Start	End	Exercise	Program	Distance or Duration	Intensity	Frequency	Duration (Pre Injection)	Duration (Post Injection)	Tumor Growth	Tumor Microenvironment								
													NK	TCD4	TCD8	Treg	B	DC	N	M	MDSC
undiff. carcinoma	[144]	Ehrlich	d0	d14	swim	yes	1 h/d	50% max. cap.	5 d/week	4 weeks	2 weeks	↓							↓	↓	
liver	[145]	Hepa1-6	d0	d42	swim	yes	5–8 min/d	variable	5 d/week	3 weeks	6 weeks	↓				↓					
breast	[146]	4T1	d0	1 cm³	run	yes	250 m/d	to 12 m/min	5 d/week	8 weeks	no	↓			↑	↓					
breast	[147]	4T1	d0	d22	run	yes	600–900 m/d	10–15 m/min	5 d/week	20 days	no	↓	↑								
breast	[147]	4T1	d0	d22	run	yes	360 m/d	6 m/min	5 d/week	20 days	no	↔									
breast	[148]	4T1.2	d0	d35	run	no	2.5–8.7 km/d	variable	daily	8 weeks	5 weeks	↔			↔						↔
breast	[149]	4T1	d0	d28	run	no	variable	variable	daily	6 weeks	4 weeks	↔									↓
breast	[150]	E0771	d0	d14	run	no	variable	variable	daily	5 weeks	2 weeks	↔	↔		↔						
breast	[83]	I3TC	d0	d32	run	no	variable	variable	daily	2 weeks	8 weeks	↓	↔	↔	↑				↔	↔	
lung	[150,151]	LLC	d0	d14	run	no	variable	variable	daily	4–5 weeks	2 weeks	↓	↑		↔	↑				↑	
melanoma	[150,151]	B16	d0	d14	run	no	4.1 km/d	variable	daily	4–5 weeks	2 weeks	↓	↑↑↑	↑	↑		↔	↑			↔
melanoma	[151]	B16	d0	d14	run	no	4.1 km/d	variable	daily	4 weeks	no	↓									

**Table 2 cancers-15-04668-t002:** Preclinical data on the exercise-induced impact on anti-tumor immunity in a therapeutic setting (=starting exercise post-tumor inoculation). Arrows upwards (↑) indicate an increase, arrows downwards (↓) indicate a decline/reduction in tumor growth or intra-tumoral immune cell numbers.

Entity	Study	Cell line	Start	End	Exercise	Program	Distance or Duration	Intensity	Frequency	Duration (Pre Injection)	Duration (Post Injection)	Tumor Growth	Tumor Microenvironment								
													NK	TCD4	TCD8	Treg	B	DC	N	M	MDSC
pancreas	[152]	KPC 4662	d0	d21	run	yes	270 m/d	9 m/min	5 d/week	no	3 weeks	↓	↑	↑	↑↑↑	↔	↓	↔/↓	↓	↔/↓	↓
breast	[154]	4T1	d0	d30	run	yes	540 m/d	18 m/min	5 d/week	no	3 weeks	↓	↑		↑						↓
breast	[153]	E0771	d0	d14	run	yes	30–45 min/d	60% Vmax	daily	no	~d10	↓	↔	↔	↑	↔		↔		↔	
breast	[153]	EMT6	d0	d14	run	yes	30–45 min/d	60% Vmax	daily	no	~d10	↓			↑						
breast	[153]	MCa-M3C	d0	d14	run	yes	30–45 min/d	60% Vmax	daily	no	~d20	↓	↔	↔	↑	↔		↔		↔	
breast	[155]	E0771	d0	d21	run	no	8 km/d	variable	daily	no	3 weeks	↔	↔		↔	↔					
melanoma	[151]	B16	d0	d14	run	no	4.1 km/d	variable	daily	no	2 weeks	↔									
melanoma	[155]	B16	d0	d17	run	no	9 km/d	variable	daily	no	17 days	↔	↔		↔	↔					

### 3.2. Transfer from the Preclinic into the Clinic 

Altogether, there is broad and consistent preclinical evidence that programmed exercise interventions can reprogram the intra-tumoral immune cell composition from pro- towards anti-tumoral (enhanced NK and CD8^+^ T cells, reduced MDSC and T_reg_) and thereby retard tumor growth. This applies to exercise both in a preventive and therapeutic setting. In contrast, voluntary exercise interventions could only induce beneficial effects in some tumor models in a preventive setting, whereas no impact has been observed in a therapeutic setting (Figure 4). 

These observations might be explained by a two-step model describing the exercise-mediated interplay between systemic and intra-tumoral immunity: Before exercise-induced redistribution of immune cells towards the tumor, exercise-induced mobilization of immune cells towards blood flow is needed [151,152] (Figure 5). Results of preclinical studies indicate that the threshold for triggering this exercise-induced leukocytosis and redistribution depends on (i) exercise-specific, (ii) host-specific and (iii) tumor-specific key factors:(i)Exercise-specific factors: EIL is mainly mediated by epinephrin [151,152]. Results of Wang et al. indicate that a minimum of exercise intensity and volume is needed for beneficial effects on intra-tumoral immune cell composition since effects were lost at small daily running distances at low velocities [147]. Taking the physiological correlation between exercise intensity, heart rate, cardiac output, and catecholamine release, this seems to be intuitive. For transfer from mouse to man, further studies must answer what exercise frequency, intensity, time, and type (FITT) is needed for measurable immunomodulatory effects. In this context, it should be considered that preclinical mouse experiments only cover aerobic exercise interventions (swimming, running) but no resistance training, leisure activities or other sports activities. However, during, e.g., resistance training, heart rate and engagement of β-adrenergic signaling are usually lower than during aerobic exercise. Especially resistance training is often beneficial for cancer patients by preventing and reversing tumor cachexia. Thus, more studies are needed to elucidate whether other training modalities, such as resistance training, could have a similar impact on anti-tumor immunity as aerobic exercise training. In terms of dose dependency, the role of exercise intensity and volume must be further analyzed. While all studies conducting programmed training reported beneficial effects on anti-tumor immunity, voluntary exercise outcomes differed widely. Since several studies could not report any impact on anti-tumor immunity by voluntary running, high exercise volume might blunt exercise-induced immunomodulatory effects. Notably, mice with unlimited access to running wheels run up to 9 km daily, often for hours, daily, for weeks and upon total exhaustion. This setting can hardly be applied to humans, especially not for cancer patients. Thus, in contrast to programmed exercise, the transfer of voluntary exercise interventions from rodents to patients is limited due to high volume. Nevertheless, it is striking that during voluntary running, some groups reported beneficial effects on anti-tumor immunity in a preventive setting but not at all in a therapeutic setting. This difference between the preventive and therapeutic settings cannot be observed for programmed exercise but has been consistently confirmed for voluntary running, even within the same tumor model and identical exercise setting [151]. An explanation for these results could be that in a therapeutic setting, exercise-induced effects are more dose-sensitive than in a preventive setting due to the different biology of tumorigenesis vs. tumor outgrowth. While tumor initiation (preventive setting) usually does not impact the host’s immunocompetency, manifest tumor growth (therapeutic setting) often leads to systemic immune alterations, possibly modulating the threshold for EIL and EIR [157].(ii)Host-specific factors: Besides exercise-specific factors, endogenous host immune status seems to be a key factor for provoking exercise-induced leukocytosis and modulating exercise-induced immune cell redistribution. Several studies have shown that exercise plus additional immunogenic stimuli such as metabolic restriction, targeted therapy or radiation could enhance anti-tumor immunity synergistically, even if exercise alone had no beneficial impact [148,158,159]. This indicates that the threshold for exercise-induced immunomodulatory effects might be lowered by co-medication, radiation or diet, all known to support antitumor immunity by different mechanisms such as elevated antigen presentation or altered metabolism. Furthermore, strain-intrinsic differences in immunity should be considered when comparing and transferring results of preclinical exercise intervention studies. The most commonly used strains for exercise intervention studies are either C57BL/6 or BALB/c mice that significantly differ in structural and functional parameters of their immune system [160].(iii)Tumor-specific factors: Exercise-induced leukocytosis is followed by myokine-mediated redistribution of immune cells towards the TME. In some cancer entities, effects are mainly dependent on CD8^+^ T cells. In others, it is mainly mediated by NK cells, which might relate to mutational burden and MHC class-I expression. In contrast to CD8^+^ T cells, NK cells preferentially recognize cells with low MHC class-I levels [161,162]. While Pedersen et al. described IL-6-induced CD8^+^ T cell enhancement as a key regulator of TME reprograming in melanoma [151], Kurz et al. claim IL-15-induced NK cell redistribution as a main factor of exercise-induced enhancement of anti-tumor immunity in pancreatic ductal adenocarcinoma [152]. Whether NK or T cells arethe main mediators, correlates with antigen presentation, a tumor intrinsic feature, rather than with the exercise program. This indicates that cancer biology (high- vs. low immunogenic) might also influence EIL and EIR.

**Figure 5 cancers-15-04668-f005:**
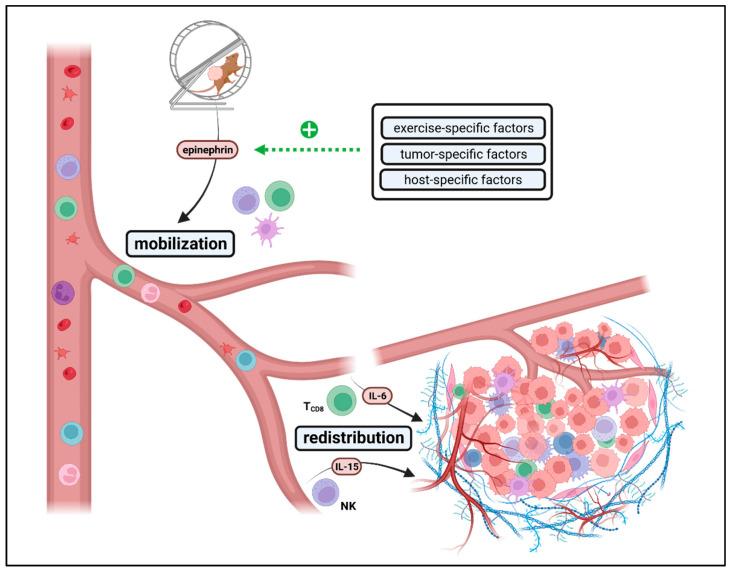
Exercise-induced interplay between systemic and intra-tumoral immunity. Exercise induces epinephrin-mediated mobilization of leukocytes (exercise-induced leukocytosis, EIL) into circulation. Those highly effective immune cells are redistributed towards peripheral tissues, such as tumors (exercise-induced redistribution, EIR). The threshold for EIL and EIR depends on exercise-, tumor- and host-specific factors. According to tumor intrinsic features and the myokine profile secreted (e.g., IL-6, IL-15), exercise-induced effects on anti-tumor immunity are mediated by different immune cell populations, such as CD8^+^ T cells or NK cells. The figure was created with https://biorender.com (accessed on 11 September 2023).

### 3.3. Clinical Data 

While most studies on exercise-induced anti-cancer effects have been reported for murine cancer models, there is growing clinical evidence for exercise as a (neo-)adjuvant therapeutic approach to enhance anti-tumor immunity. Kurz et al. included 70 patients suffering from pancreatic ductal adenocarcinoma (PDA) in a single-arm exercise intervention study (NCT02295956) [152,163,164]. Patients performed a home-based unsupervised training program including 20–30 min of aerobic activity (walking) three times a week and 30 min of resistance training (resistance bands) two times a week before surgery and concurrent with neoadjuvant chemotherapy or radio-chemotherapy. Exercise was tracked by patient-conducted training logbooks and partly, wrist-worn accelerometers. Patients who participated in pre-operative exercise revealed significantly higher numbers of tumor infiltrating CD8^+^ T cells and better overall survival (OS) than the sedentary control groups. Notably, PDA is a highly immunosuppressive cancer type characterized by poor response rates to checkpoint inhibitors [165]. 

Only recently, Durhuus et al. investigated the impact of preoperative aerobic high-intensity interval training (HIIT) four times per week on the intra-tumoral NK cell composition of 30 patients with localized prostate carcinoma [166]. Exercising patients conducted supervised training sessions on a bicycle ergometer (20–25 min, four x/week, each training consisting of four to six cycles of high-intensity interval training for one minute at 100–120% of peak power output [W_peak_] followed by three minutes of active recovery at 30% of W_peak_) for two to eight weeks preoperative of prostatectomy. Within the overall analysis, no increase in intra-tumoral NK cell infiltration by preoperative exercise could be observed. However, the exercise duration varied from two to eight weeks, resulting in large differences in total exercise dose (4–30 training sessions). When analyzing the subgroup of participants who performed at least 75% of total training sessions in five weeks, a significant increase in NK cell infiltration was observed. A positive correlation between the number of training sessions and intra-tumoral NK cell abundance could also be found. These results strengthen the hypothesis of the two-factor model of EIL and EIR described for murine exercise oncology models above (Section 3.2).

## 4. Exercise-Induced Impact on Checkpoint Inhibition

Immune checkpoint blockade has revolutionized the treatment of solid cancer. However, not all tumor types respond to ICI therapy. Growing evidence about exercise-mediated changes in intra-tumoral immune cell composition indicates that physical activity might boost the efficacy of checkpoint inhibitor therapy. 

### 4.1. Preclinical Data

B16 melanoma-bearing mice pretrained by voluntary wheel running four weeks before tumor inoculation displayed a shift of the TME from pro- towards anti-tumoral and upregulation of immune checkpoints (PD-1, PD-L1, PD-L2, CD28, B7.1, B7.2). Monotherapy with wheel running significantly reduced tumor growth. However, no additive growth inhibitory effect was observed when combined with anti-PD-1 or anti-PD-L1 treatment, maybe due to the large suppression of tumor growth by wheel running alone (−72%) [150]. A mouse model of pancreatic adenocarcinoma post-transplant programmed exercise sensitized tumors to checkpoint inhibition. While monotherapy with anti-PD-1 did not benefit, combination therapy of anti-PD-1 and exercise resulted in a significant increase of cytotoxic T cells and enhanced growth inhibition [152]. These data suggest that exercise can unlock the sensitivity of recalcitrant pancreatic tumors to PD-1 therapy by shifting the TME towards an anti-tumoral phenotype. 

A mouse model of unresectable hepatocellular carcinoma (HCC) post-transplant exercise could enhance the effect of combination therapy with anti-PD-1 and the tyrosine kinase inhibitor Lenvatinib. Interestingly, mice undergoing long-term combination therapy with anti-PD-1 and Lenvatinib developed an immunosuppressive phenotype over time (increased infiltration of T_reg_ cells, upregulation of inhibitory immune checkpoints). In contrast, the exercising group did not display this reprogramming of TME [158]. Furthermore, Gomes-Santos et al. showed that MCa-M3C breast cancer mice resistant to checkpoint inhibition could be sensitized to ICI therapy by moderate-intensity running starting post-tumor inoculation. While anti-PD-1 plus anti-CTLA-4 therapy alone did not lead to growth inhibition, the combination with exercise plus ICI therapy resulted in significantly delayed tumor growth [153]. In another breast cancer mouse model (4T1) non-sensitive to monotherapy with checkpoint inhibition, the combination of ICI plus radiation (RT) plus exercise was associated with significantly slower tumor growth compared to a dual combination of RT and anti-PD-1 [154]. However, the results of this study are limited since the intervention group of exercise plus ICI therapy is not shown for comparison. On the contrary, Buss et al. did not observe the beneficial impact of exercise on checkpoint inhibition, neither in B16-F10 melanoma nor E0771 breast cancer mice. Notably, in this study, exercise intervention was conducted voluntarily, strengthening the dose-dependency of exercise-induced immunomodulatory effects [155]. 

### 4.2. Clinical Data

Besides preclinical studies, there is also rising clinical effort to analyze the impact of exercise on immune checkpoint inhibition in patients. Exercise enhanced first-line combination treatment of Lenvatinib plus ICI (anti-PD-1) in patients with unresectable hepatocellular carcinoma by fostering a shift of TME towards anti-tumoral with improved OS, progression-free survival (PFS) and overall response rate (ORR) [158]. Active patients conducted unsupervised training sessions at least five times per week of moderate aerobic activity for >30 min or at least three days per week of vigorous aerobic activity for >30 min/d or at least three to five days a week of mixed-intensity activity for more than 30 min/d before or within one month after the initiation of combination therapy. The criteria for exercise intensity were based on the American College of Sports Medicine guidelines: Vigorous-intensity aerobic activity is defined as exercise making the patient slightly out of breath and unable to talk. Moderate activity is regarded as exercise, making the patient unable to sing but rendering talking still possible [167]. Training intensity was assessed by retrospective questionnaires. 

Several studies (Charles et al. [168], Hyatt et al. [169], Lacey et al. [170], Sportivumab study [NCT03171064]) investigated the impact of exercise on cancer patients under checkpoint inhibition in terms of physical and emotional benefits but not biological outcomes. For example, in the Sportivumab study (NCT03171064), melanoma patients under checkpoint inhibitor therapy conducted a twelve-week supervised combined resistance and endurance exercise training (60 min, two x/week). Endpoints include pain, muscle strength, cardiopulmonary fitness, physical activity behavior, depression, sleep quality, fatigue, quality of life and feasibility of exercise intervention. However, blood or tumor specimens were not investigated in terms of immune-related markers. The study is completed, but the results are not published yet.

The ERICA study (NCT04676009), a prospective monocentric, randomized controlled open-label study, investigates the acute effects of one hour exercise before application of checkpoint inhibition (pembrolizumab) plus platinum-based duplet chemotherapy in 30 NSCLC patients. [171]. Patients conducted a three-month exercise program consisting of a supervised exercise session (35 min interval training at submaximal intensity) one hour before immune-chemotherapy infusion and an unsupervised home-based walking program recorded by an activity tracker. Clinical, physical, biological, and psychosocial parameters are analysed, including immune and inflammatory biomarkers from peripheral blood samples. Results are not published yet. 

The HI AIM study (NCT04263467) investigates the impact of exercise on immune cells in 70 patients with NSCLC in a randomized controlled trial by analyzing blood samples and ultrasound-guided tumor samples of patients undergoing checkpoint inhibitor or combined checkpoint inhibitor plus chemotherapy or oncological surveillance [172]. Patients in the treatment arm performed a supervised group-based exercise training consisting of intermediate to high-intensity interval training thrice weekly for six weeks. Results are not published yet.

The EDEN study (NCT04866810) explores the impact of diet and exercise on immunotherapy and the microbiome in melanoma patients receiving checkpoint inhibitor treatment. Intervention group participants receive a plant-based, high-fiber diet and perform at least 150 min of moderate or 75 min of high-intensity exercise per week, recorded app-based by a fitness tracker. The primary outcome measure is feasibility. Secondary outcome measures are PFS, quality of life (QOL) and ORR. Results are not published yet.

The Moffitt Cancer Center initiated a randomized interventional trial (NCT05358938) investigating the impact of exercise on neoadjuvant and adjuvant immunotherapy in melanoma, cutaneous squamous cell and Merkel cell carcinoma patients. Patients in the interventional arm complete 30 min of moderate exercise on an arm ergometer, a cycle ergometer, or a treadmill before each administration of ICI across all cycles. Blood samples will be collected at baseline, post-exercise, and post-infusion on the first, third, midpoint and final infusion dates. Primary outcome measures include feasibility analysis and the impact of exercise on tumor immunological biomarkers (adjuvant setting) and pathological complete response (neoadjuvant setting). Results are not published yet.

## 5. Conclusions

The current literature provides broad evidence for beneficial exercise-induced effects on anti-tumor immunity. It supports combining immune checkpoint inhibition with exercise as a promising therapeutic approach to enhance therapy response. 

In line with results obtained from tumor mouse models, exercise has been shown to shift the intra-tumoral immune cell composition from pro- towards anti-tumoral and enhance overall survival in patients, such as in pancreatic ductal adenocarcinoma and prostate carcinoma. Furthermore, preliminary results of exercise intervention studies suppose a beneficial impact of exercise on the efficacy of immune checkpoint inhibitor therapy. However, the threshold for triggering exercise-induced immunomodulatory effects relies on exercise-, host- and tumor-specific factors. Exercise-induced mobilization and redistribution of immune cells seems to depend on cancer type, host immune status and exercise frequency, intensity, time, and type (FITT). 

Thus, to design the optimal training setup for cancer patients, further studies, especially in the clinical setting and with programmed exercise interventions, are needed to address the feasibility and role of exercise on local and systemic anti-tumor immune response in different cancer types. 

## Figures and Tables

**Figure 1 cancers-15-04668-f001:**
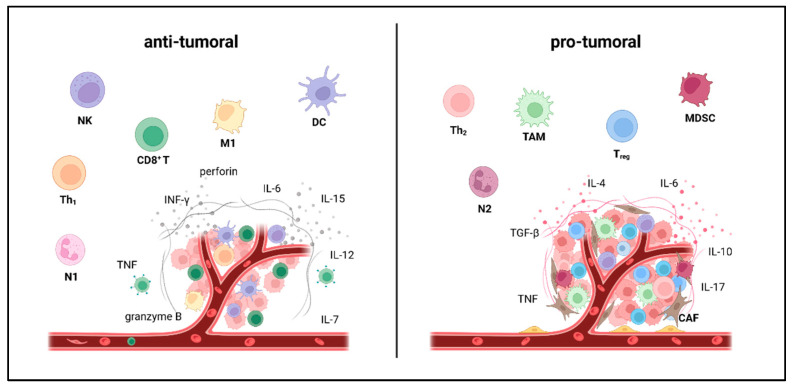
Main cellular components and driver cytokines are characteristic of an anti- vs. pro-tumoral microenvironment. Immunostimulatory cells (“anti-tumoral”) include dendritic cells (DC), CD4 positive T Helper Cells (Th_1_), cytotoxic T cells (CD8^+^ T), natural killer cells (NK), N1 neutrophils (N1) and M1 like macrophages (M1). Immune suppressive cell populations (“pro-tumoral”) include regulatory T cells (T_reg_), CD4 positive T Helper Cells (Th_2_)_,_ myeloid-derived suppressor cells (MDSC), N2 neutrophils (N2), tumor-associated macrophages (TAM) and cancer-associated fibroblasts (CAF). Shifting the immune cell composition of the tumor microenvironment from pro- towards anti-tumoral could enhance immunotherapy. The figure was created with BioRender (https://biorender.com, accessed on 11 September 2023).

**Figure 2 cancers-15-04668-f002:**
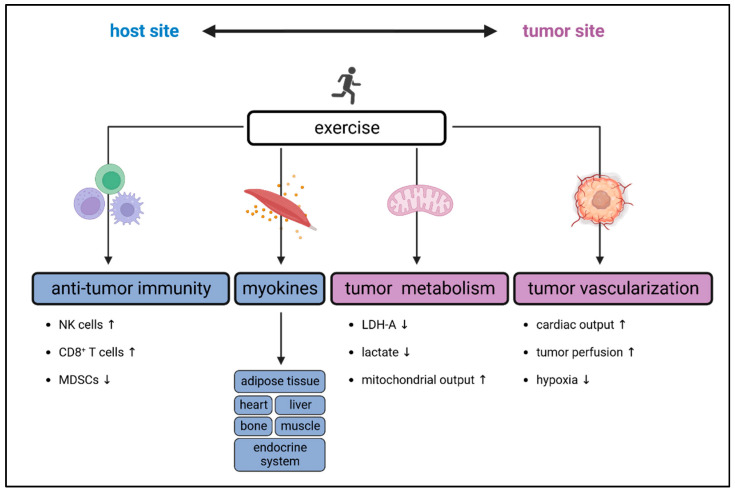
Mechanisms of exercise-induced effects on tumor biology. Exercise can impact tumor growth through an interplay of muscle-derived mediators (myokines), alterations of tumor vascularization and metabolism and changes in anti-tumor immunity. Thereby, exercise-mediated adaptions on the host- and tumor-site can influence each other (double-headed arrow). The figure was created with BioRender (https://biorender.com, accessed on 11 September 2023).

**Figure 3 cancers-15-04668-f003:**
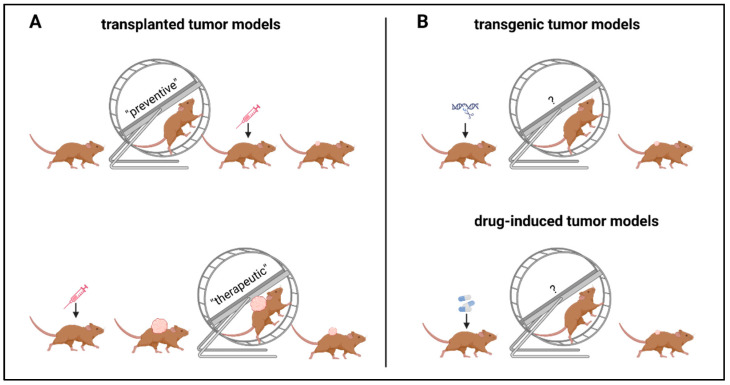
Different settings of exercise interventions in tumor mouse models. Tumors can be induced either by (**A**) transplantation or (**B**) genetic engineering and, respectively, drugs. (**A**) In transplanted models, exercise can be applied in a preventive or therapeutic setting: In the preventive setting, mice are exercised before tumor inoculation, and in the therapeutic setting, exercise is performed after tumor inoculation. (**B**) Genetic or drug-induced tumor models represent a mixed preventive and therapeutic settings model and are not included in this review. The figure was created with https://biorender.com/ (accessed on 11 September 2023).

**Figure 4 cancers-15-04668-f004:**
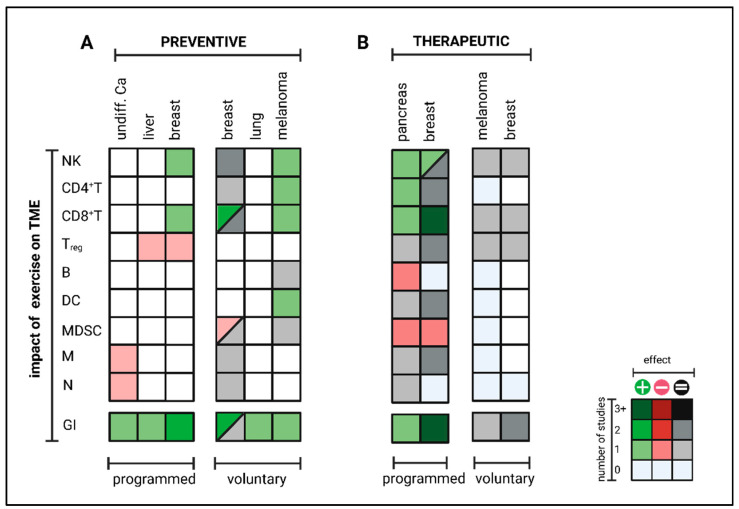
Exercise-induced impact on anti-tumor immunity in an (**A**) preventive or (**B**) therapeutic setting in mice. Mice were trained either (**A**) before (=preventive) or (**B**) post (=therapeutic) inoculation of different tumor cell lines. Growth inhibition (GI) and intra-tumoral immune cell composition, including natural killer cells (NK), T helper cells (CD4^+^ T), cytotoxic T cells (CD8^+^ T), regulatory T cells (T_reg_), B cells (**B**), dendritic cells (DC), myeloid-derived suppressor cells (MDSC), macrophages (M) and neutrophils (N) were analyzed in comparison to sedentary controls. Enhanced intra-tumoral cell numbers are given in green and reduced in red. No observed differences between exercising and sedentary groups are labelled gray, and cell subtypes not investigated are marked white. Color gradation represents the level of evidence as shown in the legend. The figure was created with https://biorender.com (accessed on 11 September 2023).

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
