# Peer review of "Can Exercise Enhance the Efficacy of Checkpoint Inhibition by Modulating Anti-Tumor Immunity?"

_cancers, 2023, doi:10.3390/cancers15184668_

Round 1
Reviewer 1 Report
The review article entitled ‘Can exercise enhance the efficacy of checkpoint inhibition by modulating anti-tumor immunity?’’ was well received.
In this paper the authors have presented literature about how does exercise impact anti-tumor immunity in a broader sense and then role in exercise in augmenting ICI therapy was described. The idea of review is interesting. The paper contains illustration where necessary. Moreover, flow of thoughts is fluent and organized. Here are few suggestions to consider to the authors.
For some set of immune cells, for example cytotoxic T lymphocytes, or CD8+ T cells. The standard way of writing is as mentioned above. In the paper it is written as TCD8. This should be addressed throughout the manuscript.
Line 46, T cells (T), no need to abbreviate. It is useless.
Line 49, macrophage (M), No need to abbreviate
Line 73, tumor associated macrophage (M2), M2, should be TAMs, as it is standard abbreviation. TAMs and M2 macrophage are synonyms not the abbreviation for one another. This should be rectified where applicable in the manuscript.
No problems
Author Response
Dear Reviewer 1,
thank you very much for reviewing this manuscript. Please find the detailed responses below and the corresponding corrections highlighted in yellow in the re-submitted manuscript.
- Comment: For some set of immune cells, for example cytotoxic T lymphocytes, or CD8+ T cells the standard way of writing is as mentioned above. In the paper it is written as TCD8. This should be addressed throughout the manuscript. Response: We agree with this comment. As suggested nomenclature of cytotoxic T cells was changed from TCD8 to CD8+ T cells throughout the manuscript.
- Comment: Line 46, T cells (T), no need to abbreviate. It is useless. Response: Agree. Abbreviation was deleted.
- Comment: Line 49, macrophage (M), No need to abbreviate. Response: Agree. Abbreviation was deleted.
- Comment: Line 73, tumor associated macrophage (M2), M2, should be TAMs, as it is standard abbreviation. TAMs and M2 macrophage are synonyms not the abbreviation for one another. This should be rectified where applicable in the manuscript. Response: Thanks for pointing this out. As suggested tumor associated macrophages are now named consistently as TAMs throughout the manuscript.
We are looking forward to your reconsideration and would like to thank you again for your time reading through the manuscript.
Yours sincerely,
Christina Brummer
Reviewer 2 Report
This review is overall good in writing and includes broad discussion from various perspectives. A few suggestions are listed below that would help to clear minor confusions.
1. In Figure 1, the N2 neutrophil should be listed in the figure legend. This would help audience to better understand what the “N” cell is in the pro-tumor panel in figure 1. In addition, there is also a subset of neutrophils (N1) that exhibit anti-tumor cytotoxicity.
2. Line 201-202. It would be good to elaborate further. While anti-inflammation might be good to interrupt tumor development, the anti-inflammatory macrophages would be also a concern of M2 macrophage generation which promotes tumor. Is there any literature to investigate macrophage polarization (M1 or M2) by exercise. There are also other subtypes of macrophages such as metabolically activated macrophages which fall in between M1 and M2.
3. Line 232. "was" provided
4. In figure 2, (a) what are the alterations of vascularization by exercise? Does exercise promote tumor angiogenesis which would promotes tumor metastasis leading to higher mortality? (b) in the tumor metabolism panel, Warburg effect demonstrates a preference for tumor cells using aerobic glycolysis to produce lactate. In the current presentation that exercise leads to TCA cycles at tumor site, does exercise reverse Warburg effect? If not, a better figure should be presented to avoid confusion.(c) does aerobic or resistance program result in the same impact on the host and tumor site? The current figure includes aerobic exercise and weight lifting.
5. Line 300, ER not HR
6. For section 3 (Line 295-300, 319-350), the 4T1 cells are mouse breast cancer cells from Balb/c and E0771 are breast cancer cells from C57Bl6. There are differences between the immune system of Balbc and B6. Thus, when reviewing literatures and comparing results from these two model systems, one should keep that in mind about their difference in immunity. Thus, different results obtained from literatures might be partly due to strain. Ex, line 319-340 used 4T1/Balbc (which has profound Th2) for programed exercise. Line 341-350 used E0771/B6 (which has profound Th1) for voluntary exercise.
7. Line 361, is there a reference to support that exercise upregulates PDL-1, PDL-2 and PD-1? These will reinforce the immune suppressive action in TME and I do not find any statement of this in ref 143.
8. The summary figure of preclinical findings in figure 4 is great and helps audience to read through.
There are a few typos that are also included in my comments above.
Author Response
Dear Reviewer 2,
thank you very much for taking the time to review our manuscript and your exceedingly helpful comments on it. We have included your suggestions as follows. Please find the corresponding corrections highlighted in blue in the re-submitted manuscript.
- Comment: In Figure 1, the N2 neutrophil should be listed in the figure legend. This would help audience to better understand what the “N” cell is in the pro-tumor panel in figure 1. In addition, there is also a subset of neutrophils (N1) that exhibit anti-tumor cytotoxicity. Response: We agree with your comment. Thus in the revised version of the manuscript we have classified neutrophils into subsets (N1 and N2). As suggested, N2 neutrophils are now listed as pro-tumoral in the figure legend (figure 1, line 107). Furthermore, N1 neutrophils were introduced as anti-tumoral and included in figure 1 and in the figure legend as well (line 105).
- Comment: Line 201-202. It would be good to elaborate further. While anti-inflammation might be good to interrupt tumor development, the anti-inflammatory macrophages would be also a concern of M2 macrophage generation which promotes tumor. Is there any literature to investigate macrophage polarization (M1 or M2) by exercise. There are also other subtypes of macrophages such as metabolically activated macrophages which fall in between M1 and M2. Response: Thanks for pointing this out. We agree that a more detailed discussion about exercise-induced effects on metabolic profiling and polarization of macrophages is needed here. Therefore, we have elucidated the impact of exercise on metabolic plasticity of macrophages in the revised manuscript (line 213-221). As suggested, we have also discussed the role of exercise on macrophage polarization in section 2.1 in brief (line 147-158). For a detailed overview the reader is further referred to the reviews of Rosa-Neto et al. (2022, ref. 79) and Callegari et al. (2023, ref. 55).
- Comment. Line 232. "was" provided. Response: Agree. We have corrected the typo (revised manuscript: line 251).
- Comment: In figure 2, (a) what are the alterations of vascularization by exercise? Does exercise promote tumor angiogenesis which would promotes tumor metastasis leading to higher mortality? (b) in the tumor metabolism panel, Warburg effect demonstrates a preference for tumor cells using aerobic glycolysis to produce lactate. In the current presentation that exercise leads to TCA cycles at tumor site, does exercise reverse Warburg effect? If not, a better figure should be presented to avoid confusion.(c) does aerobic or resistance program result in the same impact on the host and tumor site? The current figure includes aerobic exercise and weight lifting. Response: We agree that figure 2 might be difficult to understand for the reader. Thus, for more intuitive reading we have provided a more detailed figure adapted as follows (line 258-263): (a) Vascularization: Tumors display abnormal vascularization resulting in hypoxia. Exercise is known to normalize intra-tumoral blood flow by enhanced cardiac output resulting in less hypoxia and better immune cell infiltration (line 264-279). (b) Metabolism: Exercise has been shown to lower intra-tumoral lactate metabolism and therefore induce a switch towards enhanced mitochondrial output. (c) Strength vs. aerobic exercise: Most data on exercise-induced effects on anti-tumor immunity has been reported for aerobic exercise. Since preclinical and clinical data examining the impact of strength training in the context of TME reprogramming is rare or pending, we have put a more neutral icon representing exercise in the figure in the re-submitted version of the manuscript.
- Comment: Line 300, ER not HR. Response: Agree, we have corrected the typo (revised manuscript: line 320).
- Comment: For section 3 (Line 295-300, 319-350), the 4T1 cells are mouse breast cancer cells from Balb/c and E0771 are breast cancer cells from C57Bl6. There are differences between the immune system of Balbc and B6. Thus, when reviewing literatures and comparing results from these two model systems, one should keep that in mind about their difference in immunity. Thus, different results obtained from literatures might be partly due to strain. Ex, line 319-340 used 4T1/Balbc (which has profound Th2) for programed exercise. Line 341-350 used E0771/B6 (which has profound Th1) for voluntary exercise. Response: This consideration is very interesting. We have added this comment in section 3.2. and are now discussing the role of strain-intrinsic key immune features as another point to consider when transferring preclinical results into the clinic (line 513-518 in the revised manuscript).
- Comment: Line 361, is there a reference to support that exercise upregulates PDL-1, PDL-2 and PD-1? These will reinforce the immune suppressive action in TME and I do not find any statement of this in ref 143. Response: Indeed, exercise-induced upregulation of key immune regulatory molecules (e.g. PD-L1, PD-L2, PD-1) has been reported for the exercise setup and tumor mouse model used by Pedersen et al. in reference 143 (revised manuscript reference 151). However, these results are shown in a later publication of the group and are reported in detail in reference 150 (Bay et al., 2020), that we have cited elsewhere in our manuscript. We agree with your comment that citation was imprecise here and have linked the statement directly with the reference of the original data in the re-submitted manuscript (revised manuscript: line 381).
- Comment: The summary figure of preclinical findings in figure 4 is great and helps audience to read through. Response: Thanks for appreciating.
We are looking forward to your reconsideration and would like to thank you again for your time reading through the manuscript.
Yours sincerely,
Christina Brummer